# Cross-Modal Learning with Adversarial Samples

**Chao Li**[1,2]     **Cheng Deng**[1,*]     **Shangqian Gao**[2]     **De Xie**[1]     **Wei Liu**[3,*]

[1]School of Electronic Engineering, Xidian University, Xi'an, Shaanxi, China
[2]Electrical and Computer Engineering, University of Pittsburgh, Pittsburgh, PA, USA
[3]Tencent AI Lab, China
{chaolee.xd, chdeng.xd, xiede.xd}@gmail.com, shg84@pitt.edu, wl2223@columbia.edu

## Abstract

With the rapid developments of deep neural networks, numerous deep cross-modal analysis methods have been presented and are being applied in widespread real-world applications, including healthcare and safety-critical environments. However, the recent studies on robustness and stability of deep neural networks show that a microscopic modification, known as adversarial sample, which is even imperceptible to humans, can easily fool a well-performed deep neural network and brings a new obstacle to deep cross-modal correlation exploring. In this paper, we propose a novel Cross-Modal correlation Learning with Adversarial samples, namely CMLA, which for the first time presents the existence of adversarial samples in cross-modal data. Moreover, we provide a simple yet effective adversarial sample learning method, where inter- and intra- modality similarity regularizations across different modalities are simultaneously integrated into the learning of adversarial samples. Finally, our proposed CMLA is demonstrated to be highly effective in cross-modal hashing based retrieval. Extensive experiments on two cross-modal benchmark datasets show that the adversarial examples produced by our CMLA are efficient in fooling a target deep cross-modal hashing network. On the other hand, such adversarial examples can significantly strengthen the robustness of the target network by conducting an adversarial training.

## 1   Introduction

Cross-modal learning, such as cross-modal retrieval, enables a user to achieve what he/she prefers in one modality (*e.g.*, image) that is relevant to a given query in another (*e.g.*, text). However, due to the drastic growth of multimedia, learning in such tremendous amounts of multimedia data has been a new challenge. The recent success of deep learning and its role in cross-modal learning seem to obviate concerns about the performance in both accuracy and speed: exploiting deep neural networks to map data samples of different modalities into compact hash codes and using fast bitwise XOR operations to perform retrieval. Extensive efforts [31, 32, 30, 29, 39, 28, 44, 33, 23, 47, 5, 18, 21, 24, 27, 40, 12, 13, 49] have been made and achieved remarkable retrieval accuracy.

The current studies [17, 20, 37] show that deep networks are vulnerable against purposeful input samples, namely adversarial samples. These samples can easily fool a well-performed deep learning model by only adding a little perturbation which is even imperceptible to humans. Besides, adversarial samples have been observed in wide areas, such as image classification [36], object recognition [9], object detection [45], speech recognition [2], *etc*. However, the potential risks of deep neural networks being vulnerable to adversarial samples in cross-modal learning have not been delineated.

In this paper, we take cross-modal hashing retrieval as a representative case of cross-modal learning, where search space can roughly be divided into four parts: T2T, I2I, I2T/T2I, and NR, as shown in

---

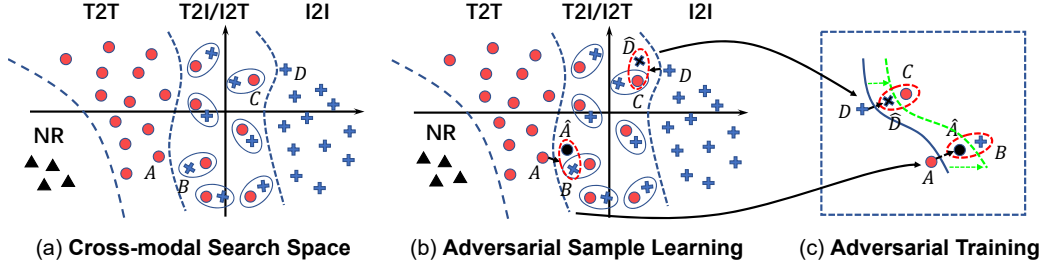

Figure 1: (a) The search space in cross-modal retrieval, which can be divided into four parts: I2I (query is image, database is image), T2T (query is text, database is text), I2T (query is image, database is text)/T2I (query is text, database is image), and NR (not relevant samples in two modalities). (b) Adversarial sample learning. (c) Adversarial training.

Fig. 1. NR means that the returned samples are not relevant to the query. In the T2T space, taking a text as a query, only semantically similar texts can be successfully retrieved, and vice versa in I2I. Different from T2T and I2I, I2T/T2I is the main concern in the cross-modal retrieval community, which focuses on information retrieval across different modalities. Existing deep cross-modal hashing works uniformly build the relationships across different modalities by constructing a multi-layer neural network, and learn hash codes via an objective of similarity metric. However, due to the lack of sufficient training data points and neglecting the robustness of their hashing networks, the search space of these methods cannot be sufficiently explored, so the retrieval performance can easily be compromised. Inspired by adversarial training [42] which successfully increases the robustness by augmenting training data with adversarial samples, we propose to explore and utilize such adversarial samples to construct a more robust search space in I2T/T2I. Different from other tasks mentioned above, for cross-modal retrieval, adversarial samples generated from four different spaces, *i.e.*, T2T, I2I, T2I/I2T, and NR, have different capacities in attacking. However, a more ideal and deceptive adversarial sample for cross-modal not only can make an effective attack to cross-modal retrieval, but also should keep the non-decreasing retrieval performance compared with a clean sample when executing single-modal retrieval. To be specific, given a cross-modal system, adversarial samples with errors in both single-modal and cross-modal are suspicious and can be easily detected. On the contrary, adversarial samples with errors merely in cross-modal but being correct in single-modal are much harder to be discovered, which are more deceptive adversarial samples.

In this paper, we propose a novel Cross-Modal Learning with Adversarial samples, namely CMLA, which can improve the robustness of cross-modal learning, such as a deep cross-modal hashing network. As such, accurate connections across different modalities can be bridged and a more robust cross-modal retrieval system can be established. The highlights of our work can be summarized as follows:

- We propose a simple yet effective cross-modal learning method by exploring cross-modal adversarial samples, where adversarial sample is defined in two aspects: two perturbations for different modalities are learned to fool a deep cross-modal network; the perturbations on each modality will not impact the performance within its modality.

- A novel cross-modal adversarial sample learning algorithm is presented. To learn cross-modal adversarial samples with high attacking capability, we propose to decrease the inter-modality similarity and simultaneously keep intra-modality similarity by one optimization.

- We additionally apply the proposed CMLA to cross-modal hash learning. Experiments on two widely used cross-modal retrieval benchmarks show the effectiveness of our CMLA in attacking a target retrieval system and further improving its robustness.

## 2 Related Works

Cross-modal hashing methods focus on building the correlation between different modalities and learning reliable hash codes, which can be mainly categorized into two settings: data-independent hashing methods and data-dependent ones. In data-independent hashing methods, hash codes are learned based on random projections, *e.g.*, locality-sensitive hashing (LSH) [16]. Compared with

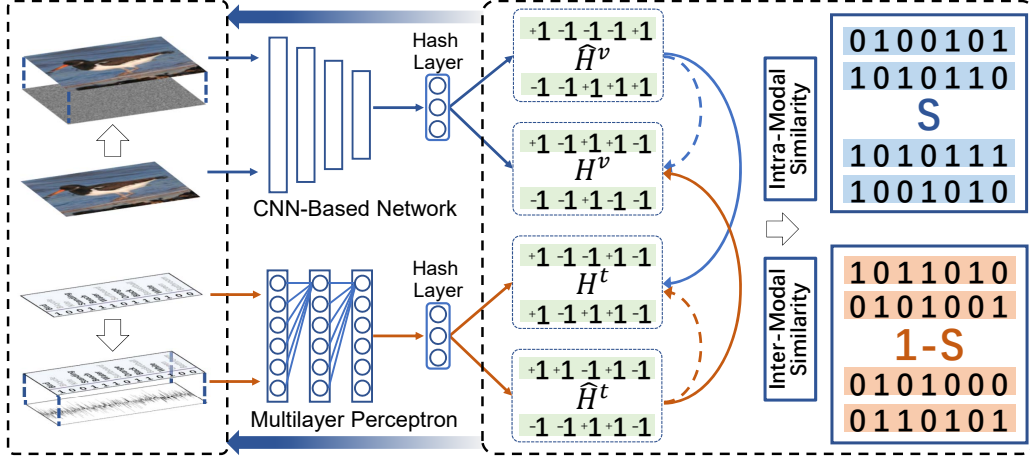

Figure 2: The pipeline of our proposed CMLA for cross-modal hash learning.

data-dependent methods, data-independent methods always require long bits to encode multi-modal data. Thus, most currently proposed methods are data-dependent ones, which can learn compact hash codes using partial data as a training set. Collective Matrix Factorization Hashing (CMFH) [14] learns hash codes for different modalities by executing collective matrix factorization of different views. Brostein *et al.* [3] presented a cross-modal hashing by preserving intra-class similarity via eigen-decomposition and boosting. Semantics-preserving hashing (SePH) [28] produces unified binary codes by modeling an affinity matrix in a probability distribution while minimizing the Kullback-Leibler divergence. Most of these methods, which depend on hand-crafted features, lack the capacity of fully exploiting heterogeneous relations across modalities. Recently, models based on deep neural networks can easily access more discriminative features, leading to a boost in the development of deep cross-modal hashing techniques [6, 15, 23, 43, 26, 11]. Deep cross-modal hashing (DCMH) [23] utilizes a deep neural network to perform feature extraction and hash code learning from scratch. Pairwise relationship guided deep hashing (PRDH) [48] integrates different pairwise constraints to purify the similarities of hash codes from inter-modality and intra-modality. Self-supervised adversarial hashing (SSAH) [25] introduces a label network into the hash learning process, which facilitates hash code generation.

Despite the outstanding performance and remarkable achievements, the DNN systems have recently been shown to be vulnerable to adversarial attacks. Szegedy *et al.* [42] first showed that a well-designed small perturbation on images can fool state-of-the-art deep neural networks with high probability. In the following, a series of research efforts on attack [41, 46, 34] and defense [7, 35, 38] have been presented. Fast gradient sign method (FGSM) [17], which uses a sign function on gradients for the inputs to learn adversarial examples, is one of the representative efficient attack algorithms. For the defense methods, adversarial training [42, 17] is adopted to augment their training data of the classifier by learning adversarial samples. Distillation technique [38] was presented to reduce the magnitude of the gradients used for adversarial sample creation and also to increase the minimum feature numbers to be modified. However, concerning large-scale cross-modal retrieval, although plenty of deep cross-modal hashing networks have been constructed, there is no attempt to focus on the security of DNNs in deep cross-modal hashing models.

## 3 Cross-Modal Learning with Adversarial Samples

### 3.1 Problem Definition

Given a cross-modal benchmark $O = \{o_i\}_{i=1}^N$ with $N$ data points, $o_i = (o_i^v, o_i^t, o_i^l)$, where $o_i^v$ and $o_i^t$ are collected in pair, respectively, denoting image and textual description for the $i$th data point, $o_i^l$ is a multi-label annotation assigned to $o_i$. Let $S$ denote a pairwise similarity matrix which describes semantic similarity between each pair of data points, where $S_{ij} = 1$ means that $o_i$ and $o_j$ are semantically similar, otherwise $S_{ij} = 0$. In a multi-label setting, when $o_i$ and $o_j$ share at

least one label, $S_{ij} = 1$; otherwise $S_{ij} = 0$. The main task of cross-modal hashing is to learn two hash functions $\mathcal{H}^*, * \in \{v, t\}$, which build cross-modal correlations and generate hash codes $B^* \in \{-1, 1\}^K$ for cross-modal data, where $K$ is code length, and $\mathcal{H}^*$ are usually learned by deep neural networks in deep cross-modal hashing. We additionally define the outputs of hash layers as $H^v$ and $H^t$ for image and text, respectively. Binary hash codes $B^*$ are generated by applying a sign function to $H^*$:

$$B^* = sign(H^*), \ * \in \{v, t\}. \tag{1}$$

For deep hashing networks $\mathcal{H}^*(o^*, \theta^*)$, let $\theta^*$ be network parameters and $J(\theta^v, \theta^t, o^v, o^t)$ be the loss function, respectively. The aim of a cross-modal adversarial attack is to find the minimum perturbations $\delta^v$ and $\delta^t$ that cause the change of retrieval accuracy. Formally,

$$\triangle(o^*, H^*) := \min_{\delta^*} \|\delta^*\|_p,$$
$$s.t. \ \max_{\delta^*} D\left(H^*\left(o^* + \delta^*; \theta^*\right), H^*\left(o^*; \theta^*\right)\right), \|\delta^*\|_p \leq \epsilon, \ * \in \{v, t\}, \tag{2}$$

where hash codes $H^*$ are generated from hash layer $\mathcal{H}^*$ by learning a deep network $\theta^*$, and $D(\cdot, \cdot)$ is a distance measure. $\|\cdot\|_p, p = \{1, 2, \infty\}$ which denotes $L_p$ norm, denotes the distance between the learned adversarial sample and the original sample.

## 3.2 Proposed CMLA

The overall flowchart of the proposed CMLA model is illustrated in Fig. 2. For a target deep cross-modal hashing network that consists of CNN-based Network and Multilayer Perceptron Network, following which are two hash layers to output hash codes for different modalities. In addition, two regularizations, namely inter-modal similarity regularization and intra-modal similarity regularization, are combined to optimize the learned adversarial samples.

For a better understanding, taking an image data point $o^v$ for example, we intend to generate an adversarial sample $\hat{o}^v$ by learning a small perturbation $\delta^v$, where $\hat{o}^v = o^v + \delta^v$. In this way, feeding $\hat{o}$ into the target deep cross-modal hashing network, semantically irrelevant results in the text modality should be returned. To achieve this goal, original text information is first fed into the deep hashing network to generate regular hash codes $H^t = \mathcal{H}^t(o^t, \theta^t)$. The correlation between the two modalities built during cross-modal hash codes generation is treated as a supervision signal to learn the optimal perturbation for each modality. With $H^t$, adversarial sample learning can be transferred to a problem of maximizing the Hamming distance between cross-modal hash codes. This can be solved by introducing an inter-modal similarity regularization, with which $\delta^v$ will be optimized by maximizing the Hamming distance between $\hat{H}^v$ and $H^t$. We formulate this inter-modal similarity loss function as:

$$\min_{\delta^v} \mathcal{J}_{inter}^v = \sum_{i,j=1}^{N} \left(\log\left(1 + e^{-\Gamma_{ij}}\right) + S_{ij}\Gamma_{ij}\right)$$
$$+ \sum_{i=1}^{N} \|\hat{o}_i^v - o_i^v\|_p, \quad \Gamma_{ij} = \frac{1}{2}(\hat{H}_i^v)(H_j^t)^\top. \tag{3}$$

Moreover, compared with single-modal retrieval, a notable difference existing in cross-modal learning is that the latter not only can build correlations across modalities but also can remain the correlations within each modality. Thus, during adversarial sample learning for cross-modal data, the modality correlation within an individual modality should be kept, which means that the learned cross-modal perturbation cannot change the intra-modal similarity relationship. To learn this perturbation, an additional intra-modal similarity regularization function is adopted in our CMLA, which can be written as:

$$\min_{\delta^v} \mathcal{J}_{intra}^v = \sum_{i,j=1}^{N} \left(\log\left(1 + e^{\Theta_{ij}}\right) - S_{ij}\Theta_{ij}\right), \quad \Theta_{ij} = \frac{1}{2}(\hat{H}_i^v)(H_j^v)^\top. \tag{4}$$

**Algorithm 1** Cross-Modal correlation Learning with Adversarial samples (**CMLA**).

---
**Input:** target deep cross-modal hashing networks: $\mathcal{H}(o^*, \theta^*), * \in \{v, t\}$, and a cross-modal dataset with $N$
       data points: {image, text, and label};
**Output:** optimal perturbations: $\delta^v, \delta^t$;
1 Maximum iteration = $T_{max}$, Batch_Size = 128, $n = \lceil N/128 \rceil$;
2 **for** $j = 1, j \leq n$ **do**
3     initialize $iter = 0$;
      **while** $iter \leq T_{max}$ **do**
4         compute
          $H^v = \mathcal{H}^v(o^v, \theta^v), \ H^t = \mathcal{H}^t(o^t, \theta^t)$;
5         **if** *not converged* **then**
6            update $\delta^v$ and $\delta^t$:
              $\delta^v = \arg\min_{\theta^v} J^v(o^v, \theta^v, H^v, H^t)$;
              $\delta^t = \arg\min_{\theta^t} J^t(o^t, \theta^t, H^v, H^t)$;
7         **end**
8     **end**
9 **end**
10 return $\delta^v$ and $\delta^t$.

---

Therefore, the objective function of our CMLA for image modality adversarial sample learning is formulated as:

$$\min_{\delta^v} \mathcal{J}^v = \alpha \sum_{i,j=1}^{N} \left( \log \left( 1 + e^{-\Gamma_{ij}} \right) + S_{ij}\Gamma_{ij} \right) + \beta \sum_{i,j=1}^{N} \left( \log \left( 1 + e^{\Theta_{ij}} \right) - S_{ij}\Theta_{ij} \right)$$
$$+ \gamma \sum_{i}^{N} \left\| \hat{o}_i^v - o_i^v \right\|_p, \tag{5}$$

where $\Gamma_{ij} = \frac{1}{2}(\hat{H}_i^v)(H_j^t)^\top$, $\Theta_{ij} = \frac{1}{2}(\hat{H}_i^v)(H_j^v)^\top$, and $\alpha$, $\beta$, and $\gamma$ are hyper-parameters. In a similar way, adversarial samples for the text modality can be learned, where the objective function is written as:

$$\min_{\delta^t} \mathcal{J}^t = \lambda \sum_{i,j=1}^{N} \left( \log \left( 1 + e^{-\Upsilon_{ij}} \right) + S_{ij}\Upsilon_{ij} \right) + \xi \sum_{i,j=1}^{N} \left( \log \left( 1 + e^{\Psi_{ij}} \right) - S_{ij}\Psi_{ij} \right)$$
$$+ \eta \sum_{i}^{N} \left\| \hat{o}_i^t - o_i^t \right\|_p, \tag{6}$$

where $\Upsilon_{ij} = \frac{1}{2}(\hat{H}_i^t)(H_j^v)^\top$, $\Psi_{ij} = \frac{1}{2}(\hat{H}_i^t)(H_j^t)^\top$, and $\lambda$, $\xi$, and $\eta$ are hyper-parameters. To solve the problems in Eqs.(5)(6), we fix the network parameters and optimize $\delta^v$ and $\delta^t$. Considering the structure difference between perturbations of image and text, CMLA learns different perturbations for two modalities, with $\delta^v$ and $\delta^t$ being updated iteratively. Algorithm 1 summarizes the learning procedure of the proposed CMLA.

## 4 Experiments

### 4.1 Experimental Setup

Extensive experiments on two benchmarks: MIRFlickr-25K [22] and NUS-WIDE [10] are conducted to evaluate the performances of our proposed CMLA and two state-of-the-art deep cross-modal hashing networks as well as their variations.

**MIRFlickr-25K** [22] is collected from Flickr, which contains 25,000 images. Each image is labeled with an associated text description. 20,015 image-text pairs are selected in our experiments, and each image-text pair is annotated with at least one of 24 unique labels. For the text modality, each text is represented by a 1,386-dimensional bag-of-words vector.

**NUS-WIDE** [10] is a public web image dataset containing 269,648 web images. 81 ground-truth concepts have been annotated for retrieval evaluation. After pruning the data point that has no label

Table 1: Comparison in terms of MAP scores of two retrieval tasks on MIRFlickr-25K and NUS-WIDE datasets with different lengths of hash codes.

| Task | Method | MIRFlickr-25K | | | | NUS-WIDE | | | |
|---|---|---|---|---|---|---|---|---|---|
| | | 16 | 32 | 48 | 64 | 16 | 32 | 48 | 64 |
| Image Query v.s. Text Database | DCMH | 0.736 | 0.749 | 0.756 | 0.761 | 0.595 | 0.607 | 0.620 | 0.641 |
| | DCMH$^+$ | 0.805 | 0.816 | 0.825 | 0.828 | 0.658 | 0.679 | 0.686 | 0.683 |
| | SSAH | 0.797 | 0.805 | 0.807 | 0.807 | 0.645 | 0.660 | 0.670 | 0.672 |
| | SSAH$^+$ | 0.804 | 0.815 | 0.826 | 0.829 | 0.660 | 0.675 | 0.690 | 0.694 |
| Text Query v.s. Image Database | DCMH | 0.796 | 0.797 | 0.804 | 0.806 | 0.601 | 0.614 | 0.623 | 0.645 |
| | DCMH$^+$ | 0.810 | 0.820 | 0.820 | 0.819 | 0.679 | 0.691 | 0.693 | 0.690 |
| | SSAH | 0.798 | 0.805 | 0.807 | 0.804 | 0.661 | 0.677 | 0.681 | 0.684 |
| | SSAH$^+$ | 0.808 | 0.809 | 0.814 | 0.815 | 0.671 | 0.685 | 0.693 | 0.697 |

or text information, a subset of 190,421 image-text pairs that belong to the 21 most-frequent concepts are selected as a dataset in our experiments. We use a 1,000-dimensional bag-of-words vector to represent each text data point.

**Evaluations.** In order to evaluate the performance of the proposed CMLA, we follow previous works [28, 4, 5] and adopt three commonly used evaluation criteria in cross-modal retrieval: Mean Average Precision (MAP) which is used to measure the accuracy of the Hamming distances, precision-recall curve (PR curve) which is used to measure the accuracy of hash lookups, and Precision@1000 curve which is used to evaluate the precision with respect to top 1,000 retrieved results. The distortion $D$ between the original modality data $o^*$ and distorted modality data $\hat{o}^*$ is measured by $D = \sqrt{\frac{\sum (\hat{o}^* - o^*)^2}{M}}, * \in \{v, t\}$. $M$ is set as 150,528 ($224 * 224 * 3$) for the image modality, while for the text modality, we set $M$ as 1,380 and 1,000 for MIRFlickr-25K and NUS-WIDE, respectively, depending on their dimensions.

**Baselines.** DCMH [23] and SSAH [25], which are two representative deep cross-modal hashing networks, are selected as the targeted cross-modal hashing models in [23] [25]. Following the setting in [23] [25], we retrain DCMH and SSAH. Moreover, in order to evaluate the cross-network transfer, we additionally construct two improved versions DCMH$^+$ and SSAH$^+$. DCMH$^+$ and SSAH$^+$ are built by replacing the vgg-f [8] network with the ResNet50 [19] network.

**Implementation Details.** Our proposed CMLA is implemented via TensorFlow [1] and is run on a server with two NVIDIA Tesla P40 GPUs holding a graphics memory capacity of 24GB for each one. All images are resized to $224 \times 224 \times 3$ before being used as the inputs. In adversarial sample learning, we use the Adam optimizer respectively with initial learning rates 0.5 and 0.002 for the image and text modalities, and train each sample for $T_{max}$ iterations. All hyper-parameters $\alpha, \beta, \lambda, \xi, \gamma$, and $\eta$ are set as 1 empirically. The mini-batch size is fixed at 128. $\epsilon^v$ is set as 8 for the image modality, and $\epsilon^t$ is set as 0.01 for the text modality. After the adversarial sample is generated, we clip the image into $0 \sim 255$ and clip text into $0 \sim 1$, respectively. The results reported in our experiments are all average results after a run for 10 times.

### 4.2 Results

For MIRFlickr-25K, 2,000 data points are randomly selected as a query set, 10,000 data points are used as a training set to train the target retrieval network model, and the remainder is kept as a retrieval database. 5,000 data points from the training set are further sampled to learn adversarial samples. For NUS-WIDE, we randomly sample 2,100 data points as a query set and 10,500 data points as a training set. Similarly, 5,000 data points from the training set are sampled to learn adversarial samples. The source codes of DCMH and SSAH are provided by the authors. Moreover, two variations DCMH$^+$ and SSAH$^+$ are constructed by replacing the vgg-f network with the ResNet50 network. All models are retrained from scratch and their performances are shown in Table 1. It can be seen that the target networks DCMH and SSAH achieve similar performances to their original papers and can achieve more promising results after equipped with ResNet50 which has more layers than vgg-f.

Given a target deep cross-modal hashing network, to evaluate the attacking performance of the proposed cross-modal adversarial samples learning method, we first fix the network parameters and

Table 2: Comparison in terms of MAP scores and distortions (D) of two retrieval tasks on MIRFlickr-25K and NUS-WIDE datasets with 32-bit code length.

| Task | Iteration | | MIRFlickr-25K | | | | NUS-WIDE | | | |
|---|---|---|---|---|---|---|---|---|---|---|
| | | | DCMH | DCMH$^+$ | SSAH | SSAH$^+$ | DCMH | DCMH$^+$ | SSAH | SSAH$^+$ |
| Image Query v.s. Text Database | 100 | MAP | 0.579 | 0.631 | 0.679 | 0.681 | 0.526 | 0.609 | 0.587 | 0.591 |
| | | D | 0.039 | 0.041 | 0.034 | 0.038 | 0.031 | 0.033 | 0.032 | 0.025 |
| | 200 | MAP | 0.563 | 0.599 | 0.671 | 0.699 | 0.499 | 0.583 | 0.534 | 0.543 |
| | | D | 0.023 | 0.038 | 0.028 | 0.032 | 0.026 | 0.031 | 0.029 | 0.026 |
| | 500 | MAP | 0.521 | 0.554 | 0.665 | 0.674 | 0.457 | 0.578 | 0.460 | 0.502 |
| | | D | 0.019 | 0.029 | 0.020 | 0.023 | 0.025 | 0.028 | 0.026 | 0.024 |
| Text Query v.s. Image Database | 100 | MAP | 0.615 | 0.619 | 0.603 | 0.611 | 0.523 | 0.628 | 0.501 | 0.523 |
| | | D | 0.048 | 0.037 | 0.031 | 0.021 | 0.037 | 0.035 | 0.042 | 0.025 |
| | 200 | MAP | 0.587 | 0.577 | 0.595 | 0.605 | 0.447 | 0.549 | 0.454 | 0.474 |
| | | D | 0.027 | 0.033 | 0.025 | 0.019 | 0.035 | 0.031 | 0.035 | 0.023 |
| | 500 | MAP | 0.561 | 0.564 | 0.589 | 0.593 | 0.371 | 0.533 | 0.351 | 0.427 |
| | | D | 0.019 | 0.021 | 0.023 | 0.017 | 0.030 | 0.027 | 0.017 | 0.019 |

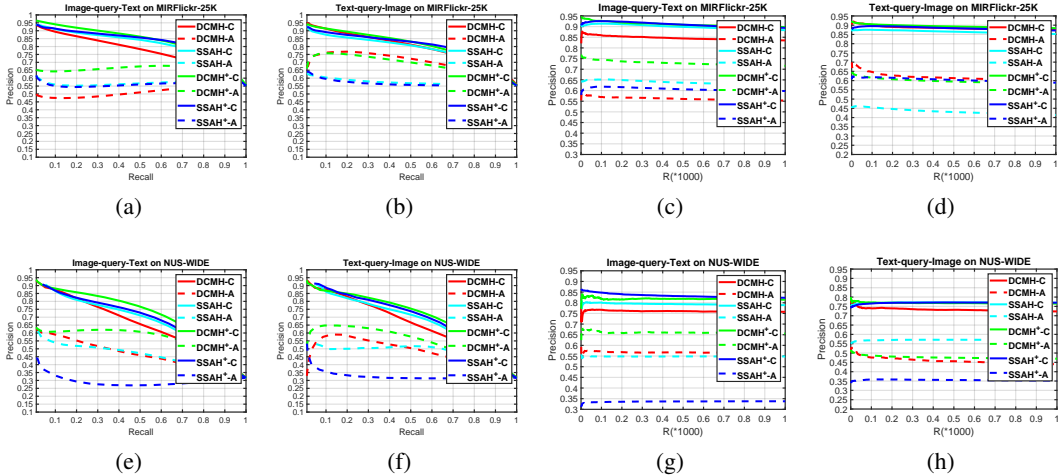

Figure 3: PR and Precision@1000 evaluated on MIRFlickr-25K and NUS-WID datasets with 32 bits.

execute CMLA to generate the adversarial sample for each query data point. Then, we respectively use each adversarial query sample to retrieve data points across different modalities, where the evaluation settings are the same as those in Table 1. Taking hash code learning with the code length of 32-bit as an example, we train each sample with different iterations from 100 to 500. The results are shown in Table 2, where we provide MAP values and distortions (D) to show the relationship between retrieval performance and distortion with the growth of training iterations. Compared with the results reported in Table 1, it is obvious that: (1) The performances of both DCMH and SSAH are severely decreased by only adding a small distortion to original modality data; (2) With the growth of training iterations, CMLA can simultaneously maintain a high attacking performance and continuously reduce the magnitude of learned disturbance. The same results are shown in Fig. 3, where PR-curves and Precision@1000 curves are provided to show the effectiveness of our proposed CMLA. Some adversarial samples and corresponding original data points are also given in Fig. 4. The images listed above are original images, where their corresponding adversarial samples are shown below them. It is nearly imperceptible to humans' eyes. For the text modality, we show the learned adversarial sample, which is constructed by mixing learned distortions and the original bag-of-words vector. Compared with the image modality, the text adversarial sample has relatively large distortions due to its nature of discrete representation.

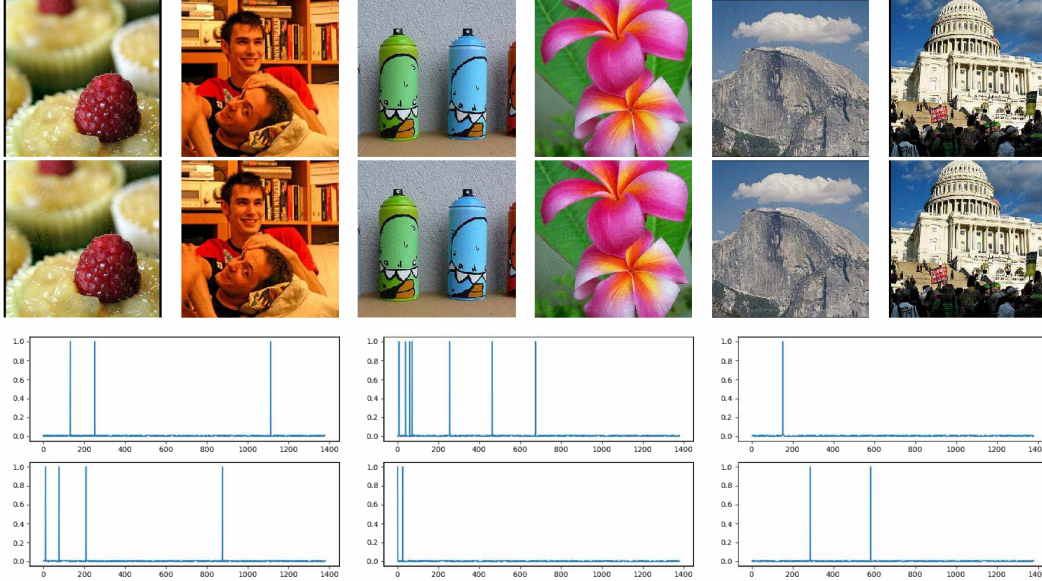

Figure 4: Adversarial samples of different modalities learned by the proposed CMLA.

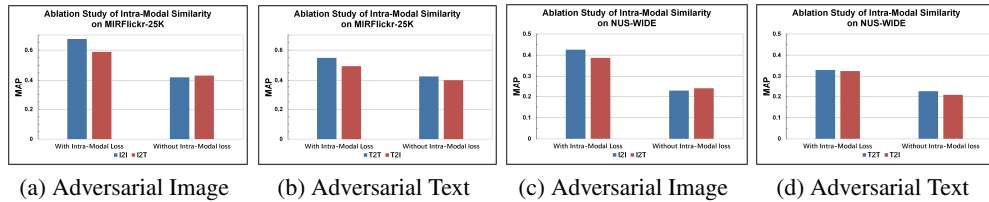

(a) Adversarial Image  (b) Adversarial Text  (c) Adversarial Image  (d) Adversarial Text

Figure 5: Ablation studies of intra-modal similarity evaluated on MIRFlickr-25K and NUS-WIDE datasets with 32 bits.

## 4.3 Further Analysis

As mentioned above, there is a main difference between cross-modal retrieval and single-modal retrieval that only retrieves data points within an identical modality. A well-designed cross-modal retrieval system can achieve both same- and different- modality data points in high accuracy. Therefore, a more deceptive cross-modal adversarial sample not only can fool a cross-modal retrieval system but also can simultaneously maintain high performance for single-modal retrieval. For a detailed elaboration, an ablation study is conducted by cutting off the intra-modal loss in our CMLA. We take SSAH as an example. Two single-modal retrieval tasks I2I and T2T are respectively executed on MIRFlickr-25K and NUS-WIDE. Fig. 5 shows the retrieval results, from which it is obvious that: The adversarial sample learned without using an intra-modal similarity constraint on MIRFlickr-25K achieves $15.7\%$ and $9.3\%$ performance drops on $I2T$ and $T2I$ compared with that learned equipped with this constraint, but the retrieval performances in single modality also obviously decrease from $0.674$ $(0.549)$ to $0.420$ $(0.426)$ on $I2I$ $(T2T)$. These adversarial samples, which have a lower performance in single-modal retrieval, can be easily detected by a single-modal retrieval verification before feeding into a cross-modal retrieval system. While equipped with our intra-modal similarity constraint, the variance of CMLA is further constrained and forced to learn more deceptive adversarial samples. In this way, CMLA can fool a cross-modal retrieval system and will not be detected by a single-modal retrieval verification.

The main goal of our adversarial sample learning is to improve the robustness of a deep cross-modal hashing network. Thus, we additionally learn adversarial samples from the training set and then conduct an adversarial training by combining the adversarial samples with the training set together. The retrieval performances of the retrained deep cross-modal hashing network under identical attacks

Table 3: Comparison in terms of MAP scores on MIRFlickr-25K and NUS-WIDE datasets with different lengths of hash codes. All networks are evaluated after adversarial training.

| Task | Method | MIRFlickr-25K | | | | NUS-WIDE | | | |
|---|---|---|---|---|---|---|---|---|---|
| | | 16 | 32 | 48 | 64 | 16 | 32 | 48 | 64 |
| Image Query v.s. Text Database | DCMH | 0.711 | 0.723 | 0.735 | 0.759 | 0.578 | 0.587 | 0.611 | 0.628 |
| | DCMH$^+$ | 0.779 | 0.781 | 0.803 | 0.801 | 0.647 | 0.649 | 0.666 | 0.677 |
| | SSAH | 0.783 | 0.784 | 0.788 | 0.785 | 0.615 | 0.640 | 0.658 | 0.661 |
| | SSAH$^+$ | 0.784 | 0.788 | 0.787 | 0.789 | 0.621 | 0.635 | 0.662 | 0.670 |
| Text Query v.s. Image Database | DCMH | 0.771 | 0.775 | 0.782 | 0.786 | 0.610 | 0.603 | 0.611 | 0.620 |
| | DCMH$^+$ | 0.793 | 0.801 | 0.803 | 0.799 | 0.655 | 0.673 | 0.675 | 0.676 |
| | SSAH | 0.790 | 0.788 | 0.792 | 0.791 | 0.638 | 0.659 | 0.660 | 0.664 |
| | SSAH$^+$ | 0.789 | 0.789 | 0.794 | 0.793 | 0.641 | 0.664 | 0.668 | 0.667 |

of adversarial query samples are shown in Table 3. Comparing Table 1, Table 2, and Table 3, we can see that each deep cross-modal hashing network achieves a significant performance increase after adversarial training. Thus, the proposed CMLA can effectively learn adversarial samples, and in turn the learned adversarial samples can also be leveraged to improve the robustness of the targeted deep cross-modal hashing network.

## 5 Conclusions

This paper presents a novel Cross-Modal Learning method with Adversarial samples, dubbed CMLA. First, we made an observation of the existence of the adversarial samples across two different modalities. Second, by simultaneously maximizing inter-modality similarity and minimizing intra-modality similarity, an effective adversarial sample learning method was proposed. Moreover, a task on cross-modal hashing retrieval was conducted to verify our proposed CMLA, where extensive results were shown in experiments. As our main purpose is to build an accurate relationship across modalities and to improve the robustness of a target retrieval system, additional adversarial training was enforced for the targeted cross-modal hashing network. The experiments on two widely-used cross-modal retrieval datasets show the high attacking efficiency of our proposed adversarial sample learning method. Besides, these adversarial samples, in turn, can further improve the robustness of existing deep cross-modal hashing networks and achieve state-of-the-art performances on cross-modal retrieval tasks.

## Acknowledgments

This work was partially supported by the National Natural Science Foundation of China 61572388, the National Key Research and Development Program of China (2017YFE0104100), and the Key R&D Program-The Key Industry Innovation Chain of Shaanxi under Grant 2018ZDXM-GY-176.

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
