[Supplementary Material]

# Cross-Modal Learning with Adversarial Samples

**Chao Li**[1,2]    **Cheng Deng**[1,*]    **Shangqian Gao**[2]    **De Xie**[1]    **Wei Liu**[3,*]
[1]School of Electronic Engineering, Xidian University, Xi'an, Shaanxi, China
[2]Electrical and Computer Engineering, University of Pittsburgh, Pittsburgh, PA, USA
[3]Tencent AI Lab, China
{chaolee.xd, chdeng.xd, xiede.xd}@gmail.com, shg84@pitt.edu, wl2223@columbia.edu

Table 1: Comparison in terms of MAP scores of two retrieval tasks between regular training and CMLA attack on IAPRTC12 with different hash code lengths.

| Task | Method | Regular Training | | CMLA Attack | |
|---|---|---|---|---|---|
| | | 16 | 32 | 16 | 32 |
| I → T | DCMH | 0.454 | 0.470 | 0.299 | 0.302 |
| | SSAH | 0.478 | 0.494 | 0.334 | 0.346 |
| T → I | DCMH | 0.480 | 0.496 | 0.298 | 0.304 |
| | SSAH | 0.488 | 0.509 | 0.289 | 0.291 |

Table 2: Comparison in terms of MAP scores of SSAH between the regular retrieval and the defense to CMLA attack under an increasing number of adversarial samples used in adversarial training on the MIRFlickr-25K dataset. Hash code length is set as 32 bits.

| Task | SSAH Adv-Trained | # Adversarial Samples Used | | | |
|---|---|---|---|---|---|
| | | 0 | 100 | 500 | 2000 |
| I → T | Regular Retrieval | 0.805 | 0.799 | 0.785 | 0.770 |
| | Defense to CMLA Attack | 0.665 | 0.681 | 0.709 | 0.784 |
| T → I | Regular Retrieval | 0.805 | 0.801 | 0.784 | 0.773 |
| | Defense to CMLA Attack | 0.589 | 0.623 | 0.667 | 0.788 |

We further evaluate our proposed CMLA on another cross-modal dataset IAPRTC12 holding richer data semantics, where 1000 and 4000 data points are respectively selected as a query set and a training set. Each text is represented as a 2912-dimensional bag-of-words vector, and each text-image pair belongs to at least one of 255 concepts. The results are shown in Table 1 on this page. The entire results have been obtained, which can again demonstrate the effectiveness of the proposed CMLA. During performing adversarial training, the robustness is about the defense to the adversarial samples. For a better illustration, we additionally evaluate our CMLA using different quantities of adversarial samples. The results are shown in Table 2 on this page. As the quantity of adversarial samples used in training increases, the performance in defense to CMLA attack also increases (robustness is increased) while the regular retrieval performance decreases. Such a trade-off is widely observed in adversarial training for regular classification tasks.

---

[*]Corresponding authors.