[Reviews · NeurIPS 2019]

Reviewer 1



This paper a new cross-modal correlation learning with adversarial samples. However, this paper still suffers from the following problem: 1. The proposed method aims to learn the minimum perturbations. A major concern is that the two perturbations for two models are learned jointly or separately. A key point for cross-modal learning are the cross-modal correlations, while the proposed scheme seems to learn the optimal perturbations separately. 2. CMLA aims to learn the perturbations which do not change intra-modal similarity relationship. An alternative way is the perturbations can be learned to enhance the robustness and stability in both the cross-modal correlations and the intra-modal similarity. It seems two adverse ends. 3. I cannot see the remarkable difference between adversarial example in the context of cross-modal and single-modal? what the unique properties of cross-modal is adopted in your method and why it is important to cross-modal analysis, esp, considering limited development in single modal adv example? In other words, it may make the paper seeming like a combination of cross-modal analysis and adversarial example, thus reducing the novelty of the paper. 4. The experiments seem insufficient to verify the effectiveness of the proposed method, especially with only two baselines and two datasets. It is expected to show the performance of other challenging cross-modal tasks. In fact, I prefer to see more experiment specifically designed for adversarial example. 5. The notations are confused and unclear, especially for the o_i in problem deļ¬nition part. In addition, though we can understand the meaning in the retrieval background, it is still expected to specify the full name of T2T, I2T, I2I, the meanings of different shapes in figure 1, and explain the NR carefully.

Reviewer 2



The paper tries to find adversarial samples in binary hashing when we have two modalities (image and text). The idea is to find the min perturbation to the image (text ) that maximizes the Hamming distance between the binary code of the perturbed image (text) and binary codes of the relevant texts (images). They show that adding these adversarial samples to the test massively decrease the performance of the network. Issues and questions: 1- My main concern is the novelty, as it is well-known that adversarial samples exist in neural network models. How does the current approach make itself different from all previous approaches in finding the adversarial samples? 2- In eq (2), the accuracy is defined based on the argmax of H(). But, H returns a d-dimensional continuous vector and its argmax does not show anything in the hashing literature. This needs more explanation. 3- In eq (3), it is clear that for S_ij=1 (similarity), the objective function finds the perturbation that maximizes the distance. But, what does happen for S_ij=1? 4- My other main concern is experiments. In table 3, the models are trained by adding adversarial samples to the training set. By comparing the results to the table 1, we can see the accuracy decreased significantly (around 4% in some cases). This is disappointing since we want to keep the accuracy as we make the network more robust. ------------------ As the authors mentioned in their response, finding adversarial examples in cross-modal learning is something new in the literature and could lead to more robust networks. I changed my score to above the threshold.

Reviewer 3



1, The idea is interesting and it works according to the experimental results. 2, The objective function in Equation5 includes some equality constraints, I was wondering how to keep this constraint during error back-propagation. And this part has to be carefully studied in the experimental section. 3. The authors stated "In this paper, we propose a novel Cross-Modal correlation Learning with Adversarial samples, namely CMLA, which, for the first time, presents the existence of adversarial samples in cross modalities data." But actually, there is some other works that also explore using adversarial loss to generate addition sample for cross-modal learning. 4. There are also many works on image and sentence matching, which are closely related to your work and should be cited and compared: 1, Look, imagine and match: Improving textual-visual cross-modal retrieval with generative models, In ICCV, 2017. 2, Learning semantic concepts and order for image and sentence matching. In CVPR, 2018. 3, Stacked cross attention for image-text matching. In ECCV, 2018.

[Author Response · NeurIPS 2019]

Table 1: Comparison in terms of MAP scores of two retrieval tasks between natural training and CMLA attack on IAPRTC12 with different hash code lengths.

| Task | Method | Natural training | | CMLA attack | |
|------|--------|------|------|------|------|
| | | 16 | 32 | 16 | 32 |
| I → T | DCMH | 0.454 | 0.470 | 0.299 | 0.302 |
| | SSAH | 0.478 | 0.494 | 0.334 | 0.346 |
| T → I | DCMH | 0.480 | 0.496 | 0.298 | 0.304 |
| | SSAH | 0.488 | 0.509 | 0.289 | 0.291 |

Table 2: Trade-off of SSAH between natural retrieval and performance under adversarial attack on MIRFlickr-25K with 32-bit hash code length.

| Task | SSAH Adv-Trained | # Adversarial Samples used | | | |
|------|------|------|------|------|------|
| | | 0 | 100 | 500 | 2000 |
| I → T | natural | 0.805 | 0.799 | 0.785 | 0.770 |
| | under attack | 0.665 | 0.681 | 0.709 | 0.784 |
| T → I | natural | 0.805 | 0.801 | 0.784 | 0.773 |
| | under attack | 0.589 | 0.623 | 0.667 | 0.788 |

1 Thank all the reviewers for their valuable comments. We have fixed all the mistakes and made responses to all questions.
2 Given your constructive suggestions, we have confidence on improving our work further.

**To Reviewer 1 :**

**1:** Considering the structure difference between image and text, CMLA learns different perturbations for two modalities, where two perturbations are updated iteratively. The correlation between different modalities is mainly learned during cross-modal hash codes generation and is then treated as a supervision signal to learn the optimal perturbation for each modality. **2:** By replacing the second term of Eq. (5) with $\sum_{i,j=1}^{n} \left\| (1 - S_{ij}) \Theta_{ij} - \log \left( 1 + e^{\Theta_{ij}} \right) \right\|^2$, CMLA can learn adversarial samples to attack both the cross-modal correlations and intra-modal similarities. However, given a cross-modal system, adversarial samples with errors in both single-modal and cross-modal are suspicious and can be easily detected. On the contrary, adversarial samples with errors merely in cross-modal but correct in single-modal are much harder to be discovered, which are more deceptive adversarial samples. This is the major reason why we want to keep intra-modal similarity. Therefore, these two types of adversarial samples are different. By utilizing adversarial samples from CMLA, the robustness of model is improved. **3:** During learning adversarial samples for a cross-modal task, the correlation between different modalities is leveraged as a guidance to generate adversarial samples with high deception. While single-modal learning only focuses on intra-modal relationship, which can be seen as a sub-problem of cross-modal counterparts. **4:** Thanks for the valuable suggestion. We further evaluate our CMLA on another cross-modal dataset IAPRTC12 holding richer data semantics, where 1000 and 4000 data points are respectively selected as a query set and a training set. Each text is represented as a 2912-dimensional bag-of-words vector, and each text-image pair belongs to at least one of 255 concepts. Due to the limited space, partial results are shown in Table 1 on this page. The entire results have been obtained and will be placed into the final version, which can again demonstrate the effectiveness of the proposed CMLA. Moreover, an additional experiment on adversarial samples is done. Please refer to our response to Reviewer 2. Other cross-modal tasks are out of the scope for this paper but can be a good guidance for future work.

**To Reviewer 2 :**

**1:** We further highlight the contributions of CMLA in the following three aspects. First, instead of simply learning adversarial samples attacking a neural network, our main contribution is to exploit adversarial samples across different modalities. Second, we simultaneously integrate inter- and intra- modality similarity regularizations across different modalities into the learning of adversarial samples, which has a great difference from a single-modal task. Finally, the task of cross-modal hashing, for the first time, is adopted to demonstrate adversarial sample learning, obviously showing the effectiveness of the proposed CMLA. **2:** Thanks for your careful checking. Following your suggestion to clearly illustrate this, we rewrite Eq. (2) as $\max_{\delta^*} D\left( H\left( x^* + \delta^*; \theta^* \right), H\left( x^*; \theta^* \right) \right), s.t. \left\| \delta^* \right\|_p \leq \epsilon, * \in \{v, t\}$, where hash codes $H^*$ are generated from hash layer $\mathcal{H}$ by learning a deep network $\theta^*$, and $D(\cdot, \cdot)$ is a distance measure. Considering the binarization of hash codes, a large divergence between $\mathcal{H}\left( x^* + \delta^*; \theta^* \right)$ and $\mathcal{H}\left( x^*; \theta^* \right)$ means a long Hamming distance between the generated hash codes, thus resulting in effective perturbations. This problem is further specified in Eq. (5), where $\max_{\delta^*} D(\cdot, \cdot)$ is equal to $\min_{\delta^*} \mathcal{J}$. **3:** In this paper, we maximize the distance for $S_{ij} = 1$, while the case of $S_{ij} = 0$ means that two data points are semantically dissimilar, so this relationship should be kept. Therefore, we don't design an individual constraint in Eq. (3). **4:** It seems that the reviewer may have misunderstood the robustness. During performing adversarial training, the robustness is about the defense to the adversarial samples. For a better illustration, we additionally evaluate our CMLA using different quantities of adversarial samples. The result is shown in Table 2 on this page. As the quantity of adversarial samples used in training increases, the performance under attacking also increases (robustness is increased) while the natural performance decreases. Such a trade-off is widely observed in adversarial training for regular classification tasks.

**To Reviewer 3 :**

**1:** Thanks for your interest in our work. **2:** In Eq. (5), the equality constraints here are just a simple replacement of $\Gamma_{ij}$ and $\Theta_{ij}$, so they do not introduce any error-prone signals. Thus, back-propagation is sufficient to optimize Eq. (5). **3:** Thanks for your valuable suggestion. Up to now, the paper referred to by the reviewer still cannot be searched. We would be glad to cite and compare this work with our CMLA in our final version if it can be totally published before that. **4:** Agree. Following your valuable suggestion, these three works will be cited to further enrich our work.

[Meta-Review · NeurIPS 2019]

There is a clear consensus among reviewers that the paper is worthy of publication. The reviewers however had some concerns on novelty. Please incorporate reviewers comment while preparing the final version.